# Deep Desulfurization of High-Sulfur Petroleum Coke via Alkali Catalytic Roasting Combined with Ultrasonic Oxidation

**DOI:** 10.3390/ma17112609

**Published:** 2024-05-28

**Authors:** Pen Luo, Zhengjie Chen, Xiuhua Chen, Wenhui Ma

**Affiliations:** 1Faculty of Metallurgical and Energy Engineering, Kunming University of Science and Technology, Kunming 650093, China; luopen510@163.com (P.L.); mwhsilicon@126.com (W.M.); 2State Key Laboratory of Complex Nonferrous Metal Resources Cleaning Utilization in Yunnan Province, Kunming University of Science and Technology, Kunming 650093, China; 3The National Engineering Laboratory for Vacuum Metallurgy, Kunming University of Science and Technology, Kunming 650093, China; 4The National Engineering Laboratory for Vacuum Metallurgy, School of Materials and Energy, Yunnan University, Kunming 650093, China; chenxh@ynu.edu.cn; 5School of Engineering, Yunnan University, Kunming 650500, China

**Keywords:** petroleum coke, desulfurization, alkali roasting, ultrasonic assisted oxidation, organic sulfur

## Abstract

The sulfur in petroleum coke is harmful to carbon products, underscoring the importance of desulfurization for high-sulfur petroleum coke. This paper proposes a method combining alkaline catalytic roasting with ultrasonic oxidation for the deep desulfurization of high-sulfur petroleum coke. The results show that the desulfurization rate reaches 88.99% and the sulfur content is reduced to 0.83 wt.% under a coke particle size of 96–75 μm, sodium-hydroxide-to-petroleum-coke ratio of 50%, roasting temperature of 700 °C, and holding time of 2 h. The alkali-calcined petroleum coke is ultrasonically oxidized and desulfurized in peracetic acid. The results show that, under a hydrogen peroxide content of 10%, hydrogen-peroxide-(liquid)-to-petroleum-coke (solid) ratio of 20 mL/g, acetic acid content of 5 mL, ultrasonic power of 300 W, reaction temperature of 60 °C, and reaction duration of 4 h, the sulfur content is reduced to 0.15 wt.% and the total desulfurization reaches 98.01%. Through a series of characterizations, the proposed desulfurization mechanism is verified. Alkali roasting effectively removes a significant portion of sulfur in petroleum coke. However, the elimination of certain sulfur compounds, such as the more complex thiophene, presents challenges. The thiophene content is subsequently removed via ultrasonic oxidation.

## 1. Introduction

Petroleum coke is a solid substance formed through the cracking of residual oil at a specific temperature and it appears as a black block with a typical amorphous carbon structure [1,2]. It is primarily composed of 85–90% carbon (C), with the remaining constituents including hydrogen (H), oxygen (O), nitrogen (N), sulfur (S), and trace amounts of metal elements. Based on the sulfur content, petroleum coke is categorized into high-sulfur coke (S > 3.0 wt.%), medium-sulfur coke (0.5 < S < 3.0 wt.%), and low-sulfur coke (S < 0.5 wt.%) [3]. Low-sulfur coke is used for the preparation of graphite electrodes [4], carburizing agents, reserve carbon [5], and some special materials [6,7,8]. Medium-sulfur coke is applied to the production of prebaking anode aluminum electrolysis, while high-sulfur coke is limited to fuel combustion in chemical plants [9,10,11], cement [12], and composite materials [13]. In recent years, the metallurgical industry has witnessed a substantial increase in the demand for petroleum coke, particularly medium-sulfur coke and low-sulfur coke. The quality of these grades directly determines the application scenarios of graphitized carbon materials derived from terminal petroleum coke. High-grade petroleum coke is extensively used in the metallurgical industry and electrode production. However, over 80% of globally produced petroleum coke falls into the low-grade category (i.e., containing 5–7 wt.% sulfur and heavy metals such as nickel and vanadium) [14], rendering it unsuitable as a conventional fuel owing to environmental concerns.

High-sulfur petroleum coke used as fuel [15] generates a large amount of harmful gases SO_x_ and NO_x_ during combustion, impacting the environment and human health [16,17]. Using high-sulfur petroleum coke as a prebaked anode material can lead to accelerated equipment corrosion, increased anode resistivity, higher power consumption of electrolytic equipment, and abnormal anode consumption owing to the presence of excessive sulfide. In the electrolytic process, sulfur is discharged in the form of SO_2_, which increases the cost of environmental protection treatment, reduces the true coke density, and increases the burn loss rate after forging, thereby leading to the cracking of the prebaked anode. In lithium anode materials based on high-sulfur petroleum coke, an excessive mass fraction of sulfur in the coke can result in defects in the lithium anode material, thereby affecting its initial coulomb efficiency [18]. During the graphitization process, an excessively high mass fraction of sulfur can disrupt the smooth operation of the graphitization device, potentially resulting in the production of a blast furnace, especially in severe cases. Therefore, to protect the environment and provide a solution to the surplus of high-sulfur petroleum coke, developing an environmentally friendly, simple, and inexpensive desulfurization technology for high-sulfur petroleum coke is crucial.

Sulfur in petroleum coke can be classified into sulfur-containing organic compounds (e.g., mercaptan, thioether, sulfone, and thiophene) and inorganic compounds (e.g., iron sulfide and sulfate). Removing inorganic sulfur is relatively straightforward, whereas organic sulfur is typically bonded to the aromatic structure on the coke’s surface, with some directly linked to the carbon chain. Furthermore, breaking the stable C–S bonds in organic sulfur poses a challenge, complicating the removal of organic sulfur. To break the C–S bond in these organic groups, various desulfurization methods for petroleum coke are employed, such as solvent extraction desulfurization, high-temperature calcination desulfurization [19,20], oxidative desulfurization [21,22], alkali metal compound desulfurization [23,24], hydrodesulfurization [25], microbial desulfurization [26], and process strengthening-assisted desulfurization [27].

High-temperature calcination is the most commonly used desulfurization method in industry settings, with the calcination temperature of industrial petroleum coke generally ranging from 1100 °C to 1400 °C [28,29,30,31] and the desulfurization rate reaching about 50% at this temperature range. High temperatures can break the C–S bond in the organic sulfur in petroleum coke to achieve desulfurization. Yu et al. [32] employed TG-MS to conduct a non-isothermal pyrolysis desulfurization experiment on petroleum coke. The desulfurization process was divided into four stages: the first stage (<500 °C), the second stage (500–900 °C), the third stage (900–1100 °C), and the fourth stage (1100–1375 °C). These stages elucidate the decomposition of various sulfur compounds in petroleum coke at different temperature intervals and the release pattern of sulfur-containing gases. However, the desulfurization effect of high-temperature calcination is related to the occurrence state of sulfide in petroleum coke, calcination technology, and the carbonaceous structure of petroleum coke. Additionally, the calcination method involves high energy consumption and yields low economic benefits.

The solvent extraction desulfurization method is based on the principle of similar phase dissolution in which various organic sulfides dissolve in selective solvents. This process effectively removes these sulfides from the coke. Ibrahim et al. [19] employed o-chlorophenol, pyridine, phenol, and furfural as solvents to remove organic sulfides from petroleum coke. Prillips et al. [33] investigated sulfur removal from petroleum coke using o-chlorophenol as an extraction agent. Agarwal et al. [34] investigated sulfur removal from petroleum coke using tetrachloroethylene and o-chlorophenol as extractants. However, solvent extraction desulfurization of high-sulfur petroleum coke tends to exhibit a low desulfurization rate, a high liquid-to-solid ratio, and an extended reaction time, leading to significant waste solvent production. This method not only increases the cost of waste solvent treatment, but also raises environmental concerns.

Microbial desulfurization is a method of removing sulfur-containing heterocyclic compounds from petroleum coke using aerobic and anaerobic bacteria under normal temperature and pressure. Microbial desulfurization has been employed to remove sulfur by releasing dibenzothiophene [35]. Microorganisms can produce dibenzothiophene in successive biochemical reactions using different enzymes through the 4S pathway. The dibenzothiophene will eventually be converted into 2-hydroxybiphenyl and sulfate. This desulfurization method retains the integrity of the carbon skeleton and the fuel calorific value of the fuel remains unchanged [36]. Agarwal. et al. [34] employed *Acidithiobacillus ferrooxidans* from Bhubaneswar, India, to study desulfurization mechanisms. The mechanism of microbial desulfurization was divided into direct oxidation and indirect oxidation. Although microbial desulfurization does not alter the bulk structure of petroleum coke, the desulfurization effect of this method is widespread, and there is a lengthy microbial culture period. The selection of specific strains is time-consuming, and the treatment of the waste solution is challenging.

Alkali metal compounds such as potassium hydroxide (KOH) and sodium hydroxide (NaOH) react with organic sulfur, sulfur salts, and sulfur-containing gases in petroleum coke at higher temperatures to effectively remove sulfur from petroleum coke [37]. Wang et al. [38] utilized NaOH to study the desulfurization performance of petroleum. The experiment, conducted under inert atmospheric conditions, involved an initial sulfur mass fraction of 6.50%, an alkali-to-petroleum-coke ratio of 50%, a temperature of 500 °C, and a reaction time of 2 h. The results indicated a 98.1% desulfurization rate of petroleum coke. Askari et al. [39] employed KOH for sulfur removal and achieved an 84.3% desulfurization rate under an initial sulfur content of 6.18%, an alkali-to-petroleum-coke ratio of 1:3, a reaction temperature of 600 °C, and a reaction time of 2 h. Similarly, Zhu et al. [40] utilized KOH and obtained a desulfurization rate of 96.3% under an initial sulfur content of 0.81% and an alkali-to-petroleum-coke ratio of 25%. Cao et al. [41] further enhanced the desulfurization effect of petroleum coke by introducing alkali in a roaster under vacuum conditions. With a roasting temperature of 1600 °C and a NaOH-to-petroleum-coke ratio of 50%, the sulfur removal rate reached 98.54%, resulting in a final sulfur content of only 0.18%.

As demonstrated in numerous desulfurization studies, a single desulfurization method is limited in the degree of sulfur removal from high-sulfur petroleum coke. Hence, in the present study, a two-stage method comprising alkali roasting and ultrasonic oxidation was employed to desulfurize high-sulfur petroleum coke. In the first stage, the high-sulfur petroleum coke raw material was roasted via alkali catalysis to remove most of the sulfur. In the second stage, the petroleum coke was treated via ultrasonic oxidation to further desulfurize the petroleum coke.

## 2. Materials and Methods

### 2.1. Materials and Characterization

The high-sulfur petroleum coke employed in this study was obtained from a factory in Baoshan City, Yunnan Province, China, and all the chemical reagents used in the experiment were analytically pure, include NaOH (FENGCHUAN, Tianjing, China), H_2_O_2_ (CHUANDONG, Chongqing, China) and acetic acid (CHRON, Chengdu, China). Petroleum coke industrial analysis was conducted according to GB/T 2001–2013 [42]. The hydrogen, oxygen, and nitrogen content of petroleum coke raw materials were determined using the hydrogen analyzer, oxygen and nitrogen analyzer (EMGA-821 and EMGA-820, HORIBA, Kyoto City, Japan, O and N sensitivity is <0.02 ppm, H sensitivity is <0.04 ppm), while the carbon and sulfur content were determined using the high-frequency infrared carbon and sulfur analyzer (HCS-140, Kaide, Shanghai, China, sensitivity is 0.1 ppm.) The results of the petroleum coke industrial analysis and elemental analysis are presented in Table 1. 

The unwashed alkali-catalyzed calcination products were characterized via X-ray diffractometry (XRD, D/Max2200PCX, Tokyo, Rigaku, Japan, the scanning speed is 3 °/min, and the scan analysis is conducted within a scanning range of 10° to 90°). The changes in functional groups and chemical bonds before and after petroleum coke treatment were characterized via Fourier-transform infrared (FTIR) spectroscopy (Thermo Nicolet iS5, Massachusetts, Thermo Fisher Scientific, Waltham, MA, USA, the spectral range is 4000 to 400 cm^−1^, with a resolution of 4 cm^−1^). The morphology of the sulfur in the petroleum coke before and after treatment was analyzed via X-ray photoelectron spectroscopy (XPS, Thermofisher ESCALAB Xi^+^, Thermo Fisher Scientific, USA, the energy range is 0~1300 eV, and the spectrum is calibrated to the C 1s peak at 284.8 eV). The structural characteristics of petroleum coke before and after treatment were analyzed using a specific surface and porosity analyzer (ASAP 2460, Micromeritics, Rockrose, GA, USA, the pore size analysis range is 0.0005 m^2^/g to no upper limit; the pore size measurement range is 3.5 A to 5000 A, and the minimum detectable pore volume is 0.0001 cc/g). The field-emission electron microscope (Nova NanoSEM 450, FEI, Portland, OR, USA, the magnification range is from 30 to 1,000,000 times, the accelerating voltage is from 200 V to 30 kV, and the electron beam current range is from 0.3 pA to 100 nA) was utilized to observe the microstructure changes in petroleum coke before and after desulfurization, as well as for scanning electron microscopy coupled with energy-dispersive X-ray spectroscopy analysis.

The sulfur content of the petroleum coke raw material was 7.54%, which decreased to 0.83% after alkali roasting. The calculation formula for the desulfurization rate *ω* is shown in Equation (1).
(1)$$\omega =\frac{S_{0}-S_{c}}{S_{0}}\times 100\%$$
where *S*_0_ is the sulfur content of petroleum coke after raw material or first-stage roasting, and *S_c_* is the sulfur content of petroleum coke under the specific condition.

### 2.2. Experimental Methods

The specific desulfurization method for petroleum coke used in this study is as follows. First, the raw petroleum coke was crushed and dried in a 105 °C blast drying oven for 24 h. Then, the large petroleum coke particles were ground using a vibration mill. Afterward, a vibrating screen was used to separate the petroleum coke powder sample into different particle sizes. Finally, the samples were dried in an oven before use. The first stage of desulfurization involved alkali roasting, in which the petroleum coke powder was uniformly mixed with alkali in an agate mortar. Subsequently, alkali roasting was conducted in the graphite boat of a tube furnace. Throughout the experiment, high-purity Ar (99.99%) was used, the flow rate was 100–150 mL/min, the temperature was 400–1000 °C, the heating rate was 5 °C/min, the alkali-to-petroleum-coke ratio was 10–70%, and the holding time was 0.5–3 h. After alkali roasting, the product was washed in a water bath at 60 °C with stirring for 30 min, then filtered by suction, and finally dried in a drying oven at 85 °C for 24 h to obtain alkali-calcined petroleum coke products.

In the second stage of desulfurization, ultrasonic oxidation desulfurization was employed to treat the petroleum coke powder after alkali roasting. The ultrasonic generator (CBS-SCL10A, GUOBIAO, Hangzhou, China) was utilized to produce ultrasonic waves. Considering that the temperature of the ultrasonic process continued to rise, the beaker was placed into a water bath to control the temperature. The quantity of petroleum coke was 4 g, H_2_O_2_ was used as the oxidant, with a concentration ranging from 2% to 14%. The H_2_O_2_-solution-(liquid)-to petroleum-coke (solid) ratio was set to between 5 mL/g and 30 mL/g, and 36% analytically pure acetic acid was used as the acid, with an addition amount of 1 mL to 10 mL. In the ultrasound-assisted treatment, the power was 100–500 W, the frequency was 19.899 kHZ, the leaching temperature was 30–80 °C, and the leaching time was 1–6 h. The oxidized petroleum coke was filtered by suction and then dried in an oven at 85 °C for 24 h to obtain a low-sulfur petroleum coke product. The experimental flow chart is illustrated in Figure 1.

## 3. Results and Discussion

### 3.1. Effect of Alkali Catalytic Roasting on Desulfurization of Petroleum Coke

The addition of alkali to petroleum coke promotes the breakage of the C–S bond, and the alkali can react with the produced hydrogen sulfide, prevent the recombination of free radicals to form more stable thiophene sulfur compounds, and promote desulfurization. The X-ray diffraction characterization of unwashed alkali-calcined products (Figure 2) revealed that they mainly consist of Na_2_S and Na_2_SO_4_. The sulfur in coke is found mainly in thiophenes (R-C_4_H_4_S), organic sulfides (R-S-R_1_), and mercaptans (R-SH), where R and R_1_ are aliphatic moieties; therefore, the first step is the thermal decomposition of the mercaptans, resulting in olefin and hydrogen sulfide. Then, H_2_S reacts instantly with NaOH, forming Na_2_S and water. The mechanism for thiophenes and organic sulfides can be considered to be alike, since the sulfur atom is bonded similarly to the molecule: -S-. The desulfurization reaction of petroleum coke after the addition of NaOH is expressed in Equations (2) and (3), and the desulfurization mechanism is shown in Figure 3.
(2)$$\text{R-SH}\;\to R\prime +H_{2}S\;;\;H_{2}S+2\;\mathrm{NaOH}\to {\mathrm{Na}}_{2}S+2\;H_{2}O$$
(3)$$7\;(\text{-S-})+6\;\mathrm{NaOH}\to 2\;{\mathrm{Na}}_{2}S+{\mathrm{Na}}_{2}{\mathrm{SO}}_{4}+3\;H_{2}S+{\mathrm{SO}}_{2}$$

#### 3.1.1. Particle Size of Petroleum Coke

The effect of petroleum coke particle size (380–250 μm, 250–180 μm, 180–150 μm, 150–96 μm, 96–75 μm, and 75–48 μm) on the desulfurization of high-sulfur petroleum coke was investigated under alkali catalytic roasting conditions of 50% NaOH and petroleum coke proportions, 600 °C roasting temperature, and 1 h holding time (Figure 4a).

Wang et al. [38] found that the desulfurization effect of petroleum coke has little correlation with the particle size of petroleum coke. However, Pintowantoro et al. [43] found that, as the particle size of petroleum coke decreased, the desulfurization effect increased, and a particle size of 100 mesh was sufficient to destroy the chemical bonds of sulfur compounds in petroleum coke.

With the decrease in petroleum coke particle size, the desulfurization rate after alkali roasting gradually increased, because the increase in the relative surface area leads to greater exposure of organic sulfur compounds, in turn increasing the likelihood of a reaction with the alkali. The desulfurization rate was 63.79% and the sulfur content was reduced to 2.73 wt.% at the petroleum coke particle size of 96–75 μm. However, when the particle size of petroleum coke exceeded 75 μm, the influence of particle size on desulfurization decreased, as the sulfur compounds were already largely exposed. The particle size of petroleum coke was determined to be 96–75 μm based on energy consumption costs.

#### 3.1.2. Alkali-to-Petroleum Coke Ratio

To investigate the influence of various alkalis and alkali contents on the desulfurization of high-sulfur petroleum coke, petroleum coke was subjected to roasting under a particle size of 96–75 μm, a roasting temperature of 600 °C, and a holding time of 1 h, as illustrated in Figure 4b. In the early stage, with the increase in the alkali content, the desulfurization efficiency gradually increased, and the petroleum-coke–alkali contact increased, resulting in a higher desulfurization rate. The addition of the alkali agent to petroleum coke can destroy the C–S bond in the organic sulfur, and the alkali agent can react with organic sulfur compounds to produce sulfur-containing gas and sulfur salts, thereby enabling desulfurization. However, the desulfurization effect of NaOH on petroleum coke varies significantly with increasing alkali content. Higher NaOH content does not necessarily enhance the desulfurization effect, mainly because the saturation of the sodium-hydroxide–sulfur interface leads to an intensified sintering reaction in the presence of excess NaOH. In the sintering process, owing to the volatile matter in the petroleum coke, the pores become blocked, inhibiting the release of sulfur-containing gas and reducing the desulfurization effect. Moreover, the sintering process may trap sulfur salts inside the petroleum coke, leading to incomplete removal of sulfur impurities during subsequent water immersion, further reducing the desulfurization effect. The morphology of petroleum coke after alkali-calcination was observed. The results indicate that petroleum coke combined with molten sodium hydroxide to form particles, and the agglomeration phenomenon became more pronounced with increasing NaOH addition. Additionally, with the increase in the potassium-hydroxide-to-petroleum-coke ratio to above 50%, the desulfurization rate of petroleum coke remained basically unchanged. Under a NaOH-to-petroleum-coke ratio of 50%, the desulfurization rate reached 63.79%. Therefore, the NaOH-to-petroleum-coke ratio was selected to be 50% for the follow-up study on the desulfurization effect of petroleum coke.

#### 3.1.3. Roasting Temperature

The effect of temperature on the desulfurization of petroleum coke was explored under a NaOH-to-petroleum-coke ratio of 50%, particle size of 96 μm to 75 μm, and holding time of 1 h. The desulfurization effect of the roasting temperature is illustrated in Figure 4c. In the early stage, with the increase in the roasting temperature, the desulfurization rate of petroleum coke products gradually increased. The reasons for desulfurization are as follows. The presence of alkali induces the side chain cracking of sulfur and aromatic hydrocarbon molecules. Owing to increased molecular activity at higher temperatures, organic sulfur in petroleum coke reacts more thoroughly with NaOH, producing more sulfur-containing gases and inorganic sulfur salts, thereby enhancing the desulfurization effect. At 700 °C, the sulfur content of petroleum coke was reduced to 2.24 wt.%, and the desulfurization rate reached 70.29%. However, beyond 700 °C, the roasting temperature adversely affected the desulfurization efficiency of petroleum coke. This is attributed to the accelerated volatilization rate of alkali with increasing temperature, resulting in reduced alkali content and incomplete reaction of organic sulfur in petroleum coke. Consequently, the desulfurization rate begins to decline. Therefore, 700 °C was determined to be the optimal temperature for alkali roasting.

#### 3.1.4. Holding Time

To explore the effect of holding time on the rate of sulfur removal from petroleum coke, the following experiments were designed to explore the influencing factors. Based on the preceding experiments, a NaOH-to-petroleum-coke ratio of 50% was adopted, the roasting temperature was set to 700 °C, and the holding time ranged from 0.5 h to 3 h. As depicted in Figure 4d, the desulfurization rate of petroleum coke increased with the increase in holding time within 2 h. This is due to the enhanced reaction between organic sulfur molecules and alkali as the holding time progresses, leading to improved desulfurization efficiency. At a holding time of 2 h, the sulfur content of petroleum coke was reduced to 0.83 wt.%, and the desulfurization rate reached 88.99%. However, beyond this duration, the holding time had a minimal impact on the purification efficiency of petroleum coke, as the reaction between sulfur and NaOH in petroleum coke was basically completed within 2 h.

### 3.2. Study on Desulfurization of Petroleum Coke by Ultrasonic-Assisted Oxidation

The sulfur content of high-sulfur petroleum coke was reduced to only 0.83% after alkali roasting with a particle size of 96–75 μm, a NaOH-to-petroleum-coke ratio of 50%, a roasting temperature of 700 °C, and a roasting time of 2 h. Because some of the more complex thiophenes, such as dibenzothiophene, cannot be pyrolyzed at this temperature, an ultrasonic oxidation treatment was conducted with respect to the petroleum coke after alkali roasting to further decrease the sulfur content.

#### 3.2.1. Hydrogen Peroxide Concentration

The influence of hydrogen peroxide (H_2_O_2_) mass fraction of 2–14% on the desulfurization of petroleum coke after alkali roasting was studied under an H_2_O_2_-solution-to-petroleum-coke ratio (liquid-to-solid ratio) of 10 mL/g, a reaction temperature of 50 °C, an ultrasonic power of 200 w, no acetic acid addition, and a leaching time of 3 h.

As depicted in Figure 5a, as the mass fraction of H_2_O_2_ increased, the content of H_2_O_2_ molecules in the oxidation system also increased, and the sulfur content of petroleum coke significantly decreased. This reduction is mainly due to the oxidation of thiophene and other sulfides in petroleum coke into soluble acids, followed by their removal via washing to achieve the desulfurization effect. In the early stage, owing to the low mass fraction of H_2_O_2_, it is insufficient, leading to an incomplete reaction with sulfide. However, as the mass fraction increases, more H_2_O_2_ molecules participate in the reaction, thereby enhancing the reaction rate and degree with sulfide. Consequently, the sulfur content in petroleum coke was significantly reduced. Up to a mass fraction of 10%, the desulfurization rate gradually increased with an increasing H_2_O_2_ mass fraction. Beyond this threshold, however, further increases in the mass fraction of H_2_O_2_ resulted in negligible changes in sulfur content, as the oxidation degree of sulfide in petroleum coke had been reached. Hence, a mass fraction of 10% for H_2_O_2_ was selected for subsequent experiments.

#### 3.2.2. Liquid-to-Solid Ratio

The effects of the ratio of H_2_O_2_ solution (liquid) to petroleum coke (solid), ranging from 5 mL/g to 30 mL/g, on the desulfurization rate of petroleum coke were studied under leaching conditions of 10% H_2_O_2_, 50 °C, 200 W ultrasonic power, no acetic acid addition, and a 3 h reaction time. As depicted in Figure 5b, the desulfurization rate of petroleum coke increased with the rise in the liquid-to-solid ratio, because, as the ratio increases, there is a greater chance for H_2_O_2_ molecules to come into contact with sulfide, leading to a more effective oxidation. At a liquid-to-solid ratio of 20 mL/g, the desulfurization rate of petroleum coke increased to 46.99%, and the sulfur content decreased to 0.44%. Further increases in the liquid-to-solid ratio had little impact on the desulfurization effect, presumably because the oxidation degree of H_2_O_2_ and sulfide in petroleum coke had essentially reached its limit under a single H_2_O_2_ oxidation system. Therefore, a ratio of 20 mL/g was selected as the optimal liquid-to-solid ratio.

#### 3.2.3. Acetic Acid Addition

The addition of acetic acid to the H_2_O_2_ solution can produce peroxyacetic acid (PAA) and, due to the instability of PAA, it can decompose hydroxyl cations (OH^+^) with a stronger oxidation capacity than H_2_O_2_. The mechanism of sulfur oxidation in petroleum coke in the CH_3_COOH–H_2_O_2_ system is shown in Equations (4) and (5). The hydroxyl cation generated by the decomposition of PAA attacks the covalent bond between the aromatic ring and the peripheral structure or the covalent bond in the aliphatic compound. This process oxidizes thiophene and sulfone into sulfuric acid, which is then removed via washing, as illustrated in Figure 6.
(4)$$CH_{3}\mathrm{COOH}+H_{2}O_{2}\to CH_{3}\mathrm{COOOH}+H_{2}O$$
(5)$$CH_{3}\mathrm{COOOH}+H^{+}⇋CH_{3}\mathrm{COOH}+{\mathrm{OH}}^{+}$$

The effect of acetic acid addition on the sulfur removal rate of petroleum coke was investigated under the following conditions: optimal H_2_O_2_ mass fraction of 10%, H_2_O_2_-(liquid)-to-petroleum-coke (solid) ratio of 20 mL/g, reaction temperature of 50 °C, ultrasonic power of 200 W, and reaction time of 3 h. As depicted in Figure 5c, in the early stage, the desulfurization rate of petroleum coke increased with acetic acid addition, because the addition of more acetic acid resulted in the generation of more PAA by H_2_O_2_; this led to the production of more hydroxyl and carboxyl radicals, resulting in enhanced oxidation effects. With the addition of 5 mL of acid, the desulfurization rate of petroleum coke reached 69.88%, and the sulfur content was reduced to 0.25%. However, with a further increase in the amount of acetic acid, the desulfurization rate began to decline. This is attributable to the excessive addition of acetic acid affecting the decomposition of PAA, thereby weakening the oxidation capacity and reducing the desulfurization rate in the system, as illustrated in Equation (5). Therefore, the recommended acetic acid addition amount is 5 mL.

#### 3.2.4. Ultrasonic Power

During ultrasonic propagation, ultrasonic “cavitation” occurs, generating shock waves and microjets when microbubbles collapse. These effects cause mechanical damage to the impurities in petroleum coke and can accelerate PAA decomposition, thereby increasing the sulfide oxidation effect of petroleum coke and reducing the sulfur content in petroleum coke.

Furthermore, the optimal conditions for the ultrasonic acid leaching of petroleum coke and the influence of ultrasonic power on the sulfur content and desulfurization rate of petroleum coke were explored. Under an optimal H_2_O_2_ mass fraction of 10%, H_2_O_2_-to-petroleum-coke ratio of 20 mL/g, acetic acid content of 5 mL, reaction temperature of 50 °C, and reaction time of 3 h, the ultrasonic power was varied from 100 W to 500 W. It can be seen from Figure 5d, with the increase in ultrasonic power, that the sulfur content of petroleum coke decreased and the desulfurization rate increased. Beyond 300 W, the desulfurization rate did not significantly change with further increases in ultrasonic power, indicating that a power level of 300 W is sufficient to enhance the oxidation system.

#### 3.2.5. Reaction Temperature

Furthermore, the influence of leaching temperature on the desulfurization rate of petroleum coke was experimentally investigated. Under an H_2_O_2_ mass fraction of 10%, H_2_O_2_-to-petroleum-coke ratio of 20 mL/g, acetic acid content of 5 mL, ultrasonic power of 300 W, and reaction time of 3 h, the reaction temperature was varied from 30 °C to 80 °C under the leaching conditions. The sulfur content and desulfurization rate of petroleum coke obtained in this temperature range are shown in Figure 5e. The desulfurization rate gradually increased with the rise in the reaction temperature in the early stage. The decomposition rate of PAA increased with increasing temperature, and the increased activity of hydroxyl cation led to higher oxidation rates of sulfur in petroleum coke, reducing the sulfur content. At a reaction temperature of 60 °C, the desulfurization rate of petroleum coke reached 77.11%, and the sulfur content decreased to 0.19%. However, beyond 60 °C, the desulfurization rate decreased with the rise in temperature. This is attributable to the accelerated decomposition rate of H_2_O_2_ at this temperature range, leading to a reduction in the molecular weight of H_2_O_2_ in the solution system. Consequently, the oxidation effect in the system weakens, resulting in incomplete oxidation of sulfide and a decrease in the desulfurization rate. Therefore, 60 °C was selected as the optimal reaction temperature.

#### 3.2.6. Comparison of the Desulfurization of Petroleum Coke via Conventional Leaching and Ultrasonic Leaching

According to the above experiments, the optimal ultrasonic-enhanced leaching conditions are as follows: an H_2_O_2_ mass fraction of 10%, H_2_O_2_-solution-(liquid)-to-petroleum-coke (solid) ratio of 20 mL/g, acetic acid addition amount of 5 mL, ultrasonic power of 300 W, and reaction temperature of 60 °C. To investigate the effect of the collaborative leaching method, the influence of non-ultrasonic conventional leaching experiments and ultrasonic-assisted leaching on the desulfurization effect of petroleum coke at different times under the same conditions was studied, and the comparison results are shown in Figure 5f.

Petroleum coke subjected to ultrasonic leaching for 4 h exhibited a desulfurization rate 38.56% higher than that of coke subjected to conventional leaching under the same conditions. This improvement is due to the synergistic effect between ultrasound and the CH_3_COOH–H_2_O_2_ system, primarily involving ultrasonic cavitation, mechanical agitation, and thermal effects. Under ultrasonic treatment, the oxidation reaction of sulfide in petroleum coke was greatly enhanced, leading to an increased removal rate of impurities. Additionally, the desulfurization rate of petroleum coke remained unchanged after 4 h of ultrasonic leaching, indicating that the sulfides susceptible to oxidation in petroleum coke are essentially fully oxidized. The application of ultrasonic technology markedly enhanced the oxidation efficiency of sulfides in petroleum coke.

### 3.3. Characterization of Petroleum Coke before and after Desulfurization

#### 3.3.1. FTIR Spectroscopy Analysis

To further elucidate the reaction process, the petroleum coke before and after treatment was analyzed via infrared spectroscopy. As shown in Figure 7, compared with the petroleum coke raw material in Figure 7a, the absorption peak of mercaptan -SH stretching vibration at 463 cm^−1^ after alkali catalytic roasting in Figure 7b basically disappeared, and the absorption peak of the C–S bond stretching vibration on the thiophene ring at 863 cm^−1^ completely disappeared. The thiophene absorption peak at 744 cm^−1^ was also significantly reduced. The C–H absorption peak of the aromatic ring at 3074 cm^−1^ disappeared completely, while the O=S=O absorption peak of the sulfone appeared at 1123 cm^−1^. It shows that most of the petroleum coke sulfur was removed after alkali catalysis, although a small part of thiophene sulfur remained and part of it was converted into sulfone sulfur. As it can be seen from Figure 7c, after the ultrasonic oxidation of calcined petroleum coke, the characteristic peak strength of thiophene at 744 cm^−1^ weakened, while that at 1123 cm^−1^ was enhanced. The results show that thiophene sulfur in petroleum coke was converted into sulfone and sulfoxide through a catalytic oxidation reaction, and part of the thiophene sulfur was oxidized to sulfate.

#### 3.3.2. XPS Analysis

Although XPS analysis is inadequate for quantifying organic sulfur in petroleum coke, it can help clarify the main forms of organic sulfur in petroleum coke [44,45]. According to reference XPS data [46,47,48,49], sulfur binding energies at the S2p peak correspond to different sulfur species: 158.7–159.6 eV for pyrite sulfur, 161.2–163.6 eV for mercaptan or thioether, 164.0–165.0 eV for thiophene sulfur, 165.0–168.3 eV for sulfone (sulfoxide) sulfur, and 168.4–175.0 eV for inorganic sulfur.

Figure 8 depicts the analysis and fitting of XPS full spectrum and S2p of petroleum coke at various treatment stages. In Figure 8a, the XPS of petroleum coke shows a prominent S peak, with an atomic proportion as high as 5.37%; however, after alkali roasting (Figure 8c), the intensity of the S peak significantly decreased, indicating that alkali addition had a significant desulfurization effect on petroleum coke. After the alkali-calcination of the petroleum coke and then ultrasonic oxidation (Figure 8e), the S peak disappeared in the XPS full spectrum, while the O peak intensity increased, indicating that the petroleum coke sulfur was removed via oxidation at this stage. As revealed by the S2p analysis and fitting of petroleum coke raw materials (Figure 8b), sulfur in petroleum coke raw materials mainly exists in the form of mercaptan (thioether) and thiophene sulfur. Moreover, as shown by the S2p analysis and fitting of petroleum coke after alkali roasting (Figure 8d), the form of sulfur changed, and sulfur compounds were affected by alkali and high temperatures. Some sulfur compounds were removed through washing, forming inorganic sulfur salts via sulfur-containing gases and alkalis, while others were converted into more stable forms such as thiophene or sulfone sulfur. As shown by the S2p analysis (Figure 8f), after ultrasonic oxidation of alkali burned petroleum coke, thiophene and mercaptan were oxidized by PAA to sulfone and then further oxidized to sulfuric acid and removed by water washing.

#### 3.3.3. Textural Properties Analysis

The structural characteristics of petroleum coke before and after treatment were characterized through the N_2_ adsorption/desorption method (Figure 9). At relative pressures higher than 0.4, the petroleum coke samples exhibited type III shape isotherms [50]. The non-closure of the absorption and desorption curves in the figure is attributable to the incomplete carbonization of petroleum coke and the process of functional group decomposition. However, the petroleum coke subjected to alkali roasting (Figure 9b) and ultrasonic oxidation treatment (Figure 9c) showed an H_4_-type hysteresis loop, indicating that the treated petroleum coke possessed a mesoporous structure. Moreover, the hysteresis loop of petroleum coke after ultrasonic oxidation was much larger than that after alkali roasting, indicating a higher porous structure formation. The surface area (S_BET_) of the sample was analyzed according to the isotherm in Figure 9d, and the total pore volume was estimated through the Barrett–Joyner–Halenda method (Figure 9e). The treated petroleum coke exhibited significantly larger specific surface area and total pore volume than the raw petroleum coke, indicating that the treatment had enhanced the porous structure. Moreover, ultrasonic oxidation resulted in a further increase in the specific surface area and total pore volume of the alkali-calcined petroleum coke, attributable to the effects of ultrasound and PAA.

#### 3.3.4. Micromorphology and SEM–EDS Analysis

The SEM image of high-sulfur petroleum coke raw material (Figure 10a) shows a flat particle structure. After alkali roasting (Figure 10b), the surface of petroleum coke became loose with small particles attached, forming a local pore structure. The results indicate that the surface morphology of the petroleum coke changed owing to the desulfurization reaction between the petroleum coke and NaOH and etching on the petroleum coke surface during calcination. The pores are attributable to the volatilization of sulfur-containing organic matter in the petroleum coke at high temperatures. Figure 10c shows the morphology of petroleum coke after ultrasonic oxidation desulfurization. The surface morphology became rougher, owing to the effects of ultrasound and PAA.

The SEM–EDS analysis of petroleum coke at different stages revealed the presence of sulfur on the surface of the petroleum coke raw material. In Figure 10d, the EDS spectrum shows a high-intensity S peak, with a content as high as 8.4%. Figure 10e shows that, after alkali catalytic roasting, the presence of sulfur was significantly reduced to 3.0%. Figure 10f shows that, following ultrasonic oxidation of the calcined petroleum coke, the sulfur content was further reduced to only 0.8%; this is highly consistent with the sulfur test results.

## 4. Conclusions

A two-stage desulphurization method characterized by a low cost, low roasting temperature, and a simple process for desulphurization of high-sulfur petroleum coke is proposed in this paper, and theoretical analyses were conducted. Under a particle size of 96–75 μm, NaOH-to-petroleum-coke ratio of 50%, roasting temperature of 700 °C, and holding time of 2 h, the desulfurization rate of high-sulfur petroleum coke reached 88.99%, and the sulfur content was reduced to 0.83 wt.%. After alkali-calcination, the petroleum coke was ultrasonically oxidized and desulfurized in PAA. At an H_2_O_2_ mass fraction of 10%, H_2_O_2_-(liquid)-to-petroleum-coke (solid) ratio of 20 mL/g, acetic acid content of 5 mL, ultrasonic power of 300 W, reaction temperature of 60 °C, and reaction duration of 4 h, the sulfur content was reduced to 0.15 wt.%. The total desulfurization efficiency of the two-stage process reached 98.01%. The petroleum coke obtained from the different stages was characterized via FTIR spectroscopy, XPS, and SEM–EDS. The results show that most of the sulfur in petroleum coke was removed via alkali roasting, but complex thiophene was difficult to remove. Through ultrasonic oxidation, it was oxidized to a soluble acid for removal. Finally, the sulfur content of petroleum coke products met the requirements of most carbon materials, and other properties of it should be tested in subsequent studies.

## Figures and Tables

**Figure 1 materials-17-02609-f001:**
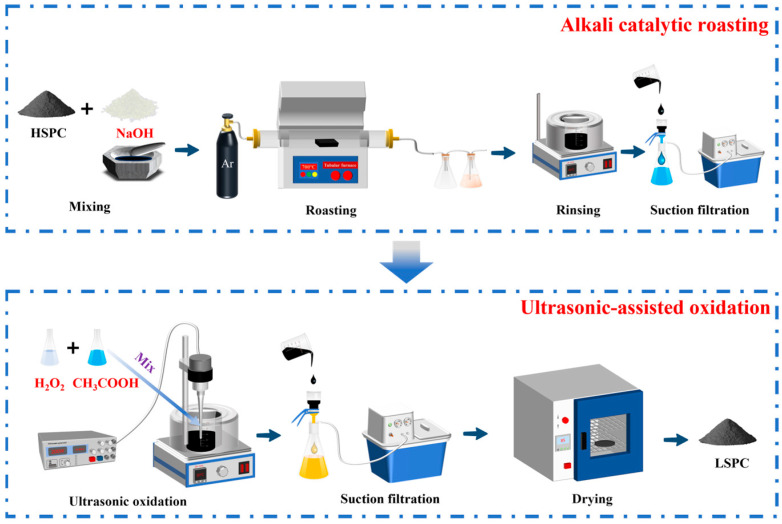
Flow chart of alkali catalyzed roasting combined with ultrasonic-assisted chemical oxidation. High-sulfur petroleum coke (HSPC) and low-sulfur petroleum coke (LSPC).

**Figure 2 materials-17-02609-f002:**
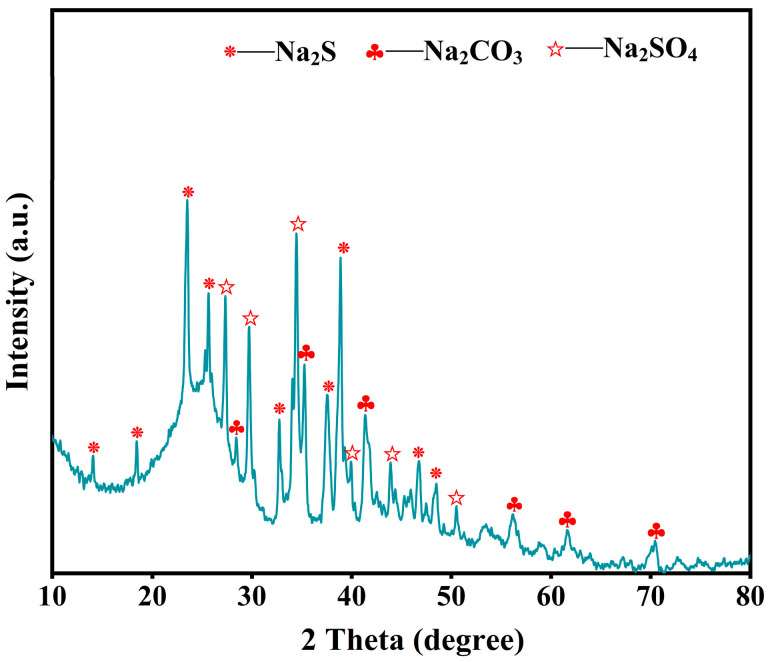
XRD pattern of unwashed petroleum coke after alkali roasting.

**Figure 3 materials-17-02609-f003:**
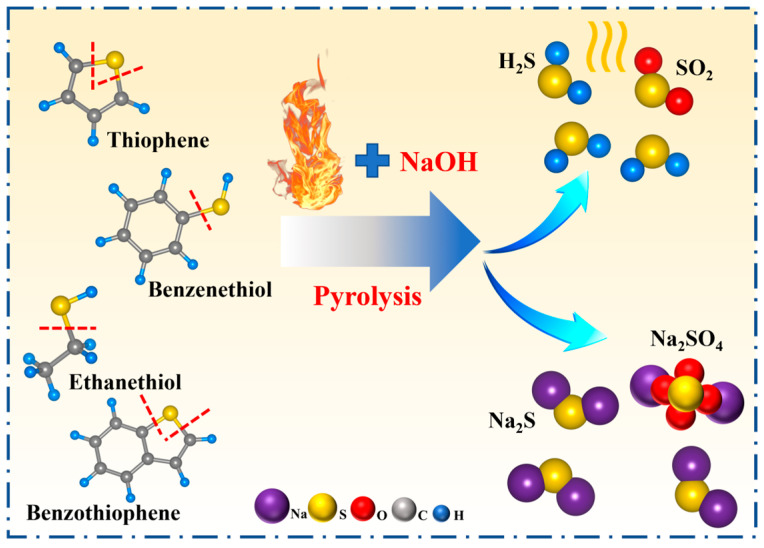
Pyrolysis mechanism of sulfur molecules in petroleum coke during the alkali roasting process.

**Figure 4 materials-17-02609-f004:**
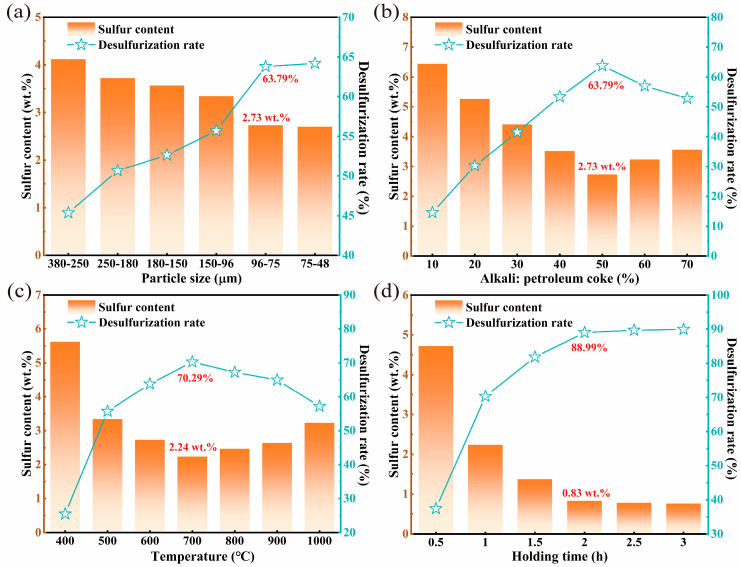
Roasting conditions of alkali catalytic roasting, (**a**) petroleum coke particle size, (**b**) alkali-to-petroleum coke ratio, (**c**) roasting temperature, (**d**) holding time.

**Figure 5 materials-17-02609-f005:**
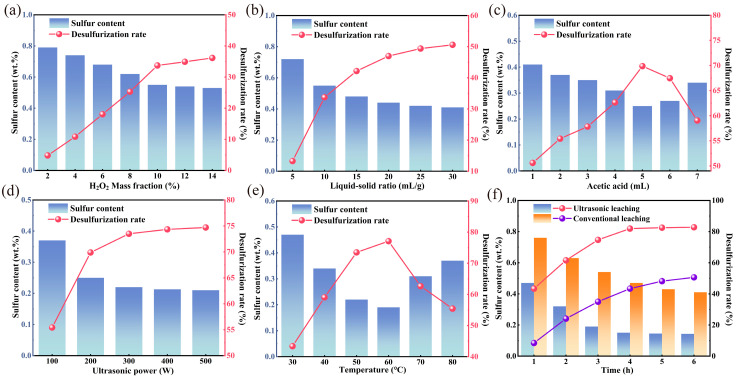
Effect of ultrasonic oxidation conditions on the desulfurization of petroleum coke: (**a**) H_2_O_2_ mass fraction, (**b**) ratio of H_2_O_2_ solution (liquid) to petroleum coke (solid), (**c**) amount of acetic acid added, (**d**) ultrasonic power, (**e**) reaction temperature, (**f**) comparison of conventional and ultrasonic leaching.

**Figure 6 materials-17-02609-f006:**
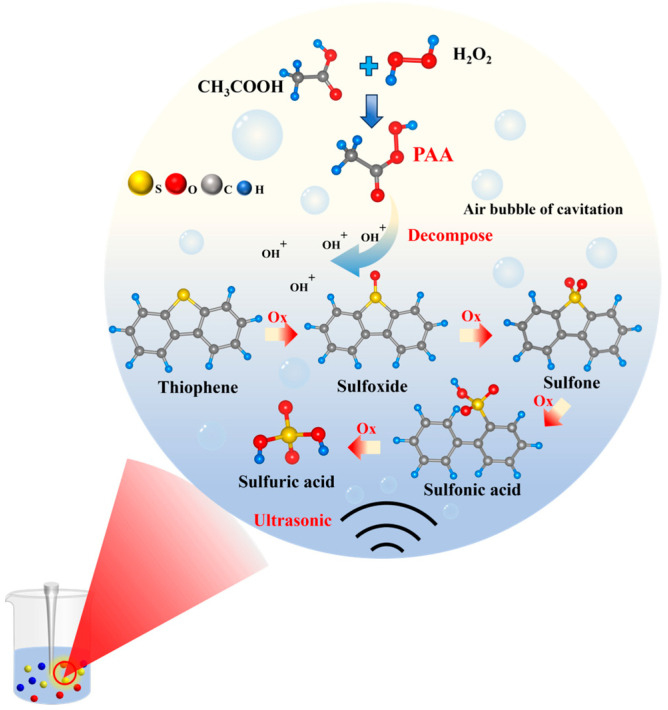
Mechanism of PAA oxidation (Ox) assisted by ultrasound.

**Figure 7 materials-17-02609-f007:**
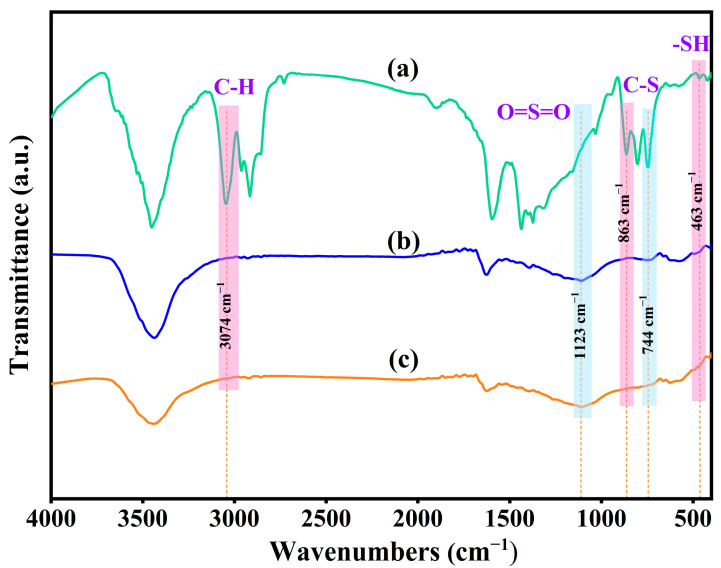
Petroleum coke FTIR images at different treatment stages. (**a**) Petroleum coke raw material, (**b**) after alkali roasting, (**c**) after ultrasonic oxidation.

**Figure 8 materials-17-02609-f008:**
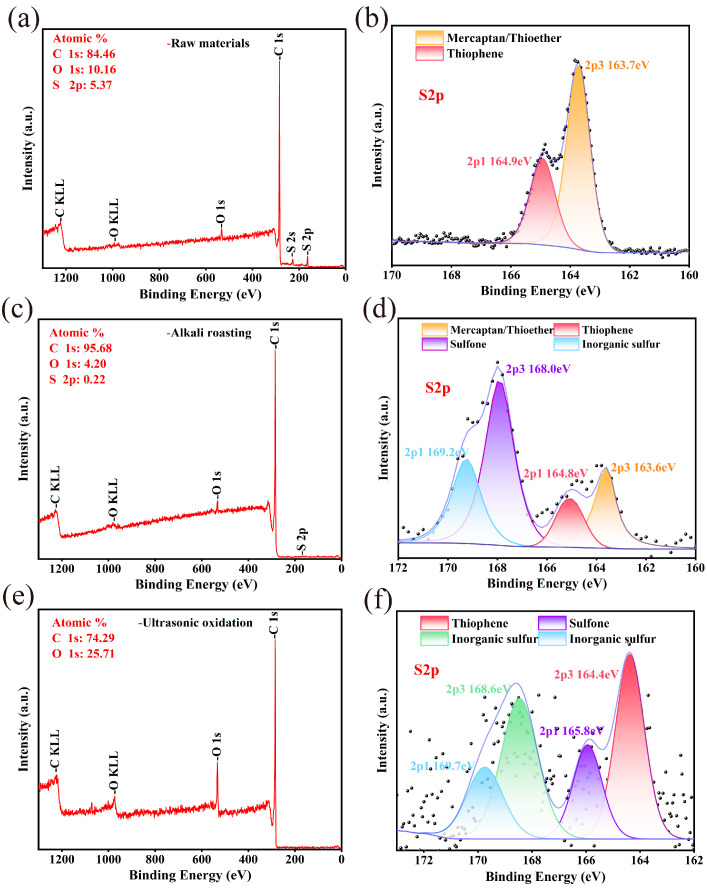
Analysis of XPS full spectrum and S2p of petroleum coke at different stages: (**a**,**b**) petroleum coke raw material, (**c**,**d**) after alkali roasting, (**e**,**f**) after ultrasonic oxidation, black dots are raw data.

**Figure 9 materials-17-02609-f009:**
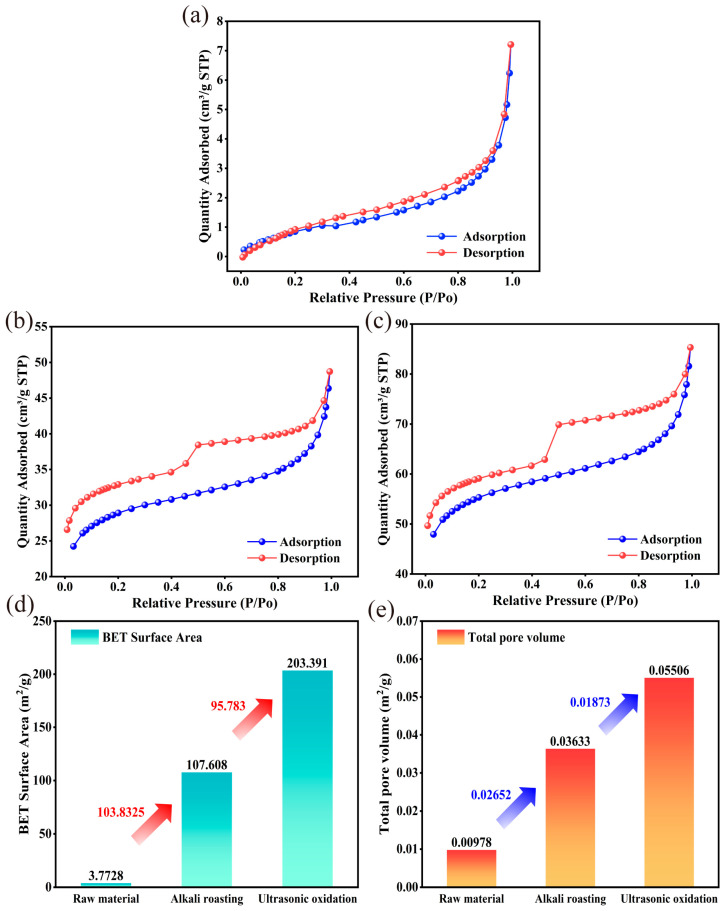
N_2_ adsorption/desorption isotherms: (**a**) petroleum coke raw material, (**b**) after alkali roasting, (**c**) after ultrasonic oxidation, (**d**) surface area, and (**e**) total pore volume of petroleum coke at different stages.

**Figure 10 materials-17-02609-f010:**
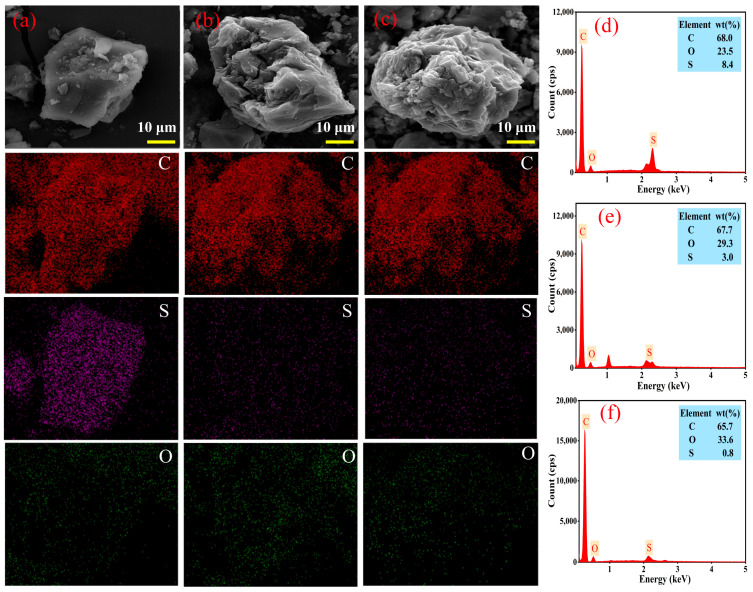
Micromorphology and SEM–EDS images of petroleum coke before and after desulfurization: (**a**,**d**) petroleum coke raw material, (**b**,**e**) after alkali roasting, (**c**,**f**) after ultrasonic oxidation.

**Table 1 materials-17-02609-t001:** Petroleum coke raw material industrial analysis and elemental analysis.

Industrial Analysis (wt.%)					Elemental Analysis (wt.%)				
M_ad_	A_ad_	V_ad_	FC_ad_	Qcal/g	C	H	O	N	S
0.36	0.38	9.9	89.36	8173.6	86.5	2.8	1.4	1.2	7.54

M_ad_—(air dry moisture); A_ad_—(air drying ash); V_ad_—(air drying volatiles); FC_ad_—(air drying fixes carbon); Q—(calorific value).

## Data Availability

The original contributions presented in the study are included in the article, further inquiries can be directed to the corresponding author.

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
