# Peer review of "Deep Desulfurization of High-Sulfur Petroleum Coke via Alkali Catalytic Roasting Combined with Ultrasonic Oxidation"

_materials, 2024, doi:10.3390/ma17112609_

Round 1
Reviewer 1 Report
Comments and Suggestions for Authors
The manuscript has been prepared well by the authors. The topic is of high interest. Although the authors have presented well, few of the suggestions are given below:
[Line 111]: A reference is usually placed before a dot.
[Line 147]: It is preferable to specify the accuracy of the equipment used.
[Line 149]: It is preferable to specify the accuracy of the equipment used.
[Line 160]: It is preferable to specify the accuracy of the equipment used.
[Line 177]: The numbering of Chapter 2.1 is already repeated [Line 142].
[Line 219]: Image title moved to another page.
[Line 223/224]: There is no space between the image title and the text.
[Line 387]: Italics are applied to the numbering.
Text in Fig. 8 and 10 is too small to understand image content. Adjustments would be needed.
[Line 510]: Is the sentence complete because there is no dot at the end of it?
[Conclusions]: It would also be necessary to outline future research in the conclusions.
[Conclusions]: Please emphasize the novelty of your findings compared to the works of other authors.
Comments on the Quality of English LanguageMinor editing of English language required.
Author Response
Dear reviewer,
Thank you for your suggestions on the article. I have answered your questions below:
Response 1: Thank you for your suggestion. The reference has been modified according to your request.
Response 2: The device information will be completed according to your requirements.
Response 3: The model and accuracy of the carbon sulfur analyzer have been completed according to your requirements.
Response 4: Thank you for pointing out the issue. The instrument accuracy has been updated.
Response 5: Thank you for pointing out the issue. The article number has been changed.
Response 6: Thank you for your suggestion. I have made the changes as per your request.
Response 7: The space has been left according to your preference.
Response 8: Thank you for your find, the bolded issue number has been fixed.
Response 9: Thank you for pointing out the issue. I have fulfilled your request by enlarging Fig. 8 and 10.
Response 10: Thank you for pointing out the issue, the punctuation has been added.
Response 11: Thank you for your suggestions. The future research direction has been outlined in the conclusion as you requested.
Response 12: Thank you for your suggestions. The advantages of this paper and other studies have been highlighted in the conclusion.
Please refer to the attachment to revise the position of the latest manuscript, thank you!

Reviewer 2 Report
Comments and Suggestions for Authors
The article investigates a highly interesting topic, which is adequately described in the introduction and satisfactorily deepened in the experimental section. The discussion is well supported from the results and the final conclusions are coherent with the results achieved.
I have only some minor comments to consider during revision:
Above the title, select the "Type of the paper".
Keywords should appear attached to the Abstract.
Line 36: the percentage indicated in the text should be specified (% in weight).
Remove the sentence at Line 151, it belongs to the Journal's template.
Line 213: revise the punctuation.
Revise the bibliography according to the required format.
Comments on the Quality of English LanguageThe english language is suitable for publication.
Author Response
Dear reviewer,
Thank you for your comments on the article, the responses to your questions are posted below
Response 1: Thank you for pointing out the question, the "Type of the Paper (Article) "has been added.
Response 2: Thank you for pointing out the problem, the key words have been added to the end of the abstract.
Response 3: Thank you for your suggestion, "wt%" has been added in the article.
Response 4: Thank you for pointing out the problem, this paragraph has been deleted from the article.
Response 5: Thank you for pointing out the problem, the punctuation has been changed in the article.
Response 6: Thank you for your suggestions. The reference format of the article has been modified.
Response 1: Thank you for pointing out the question, the "Type of the Paper (Article) "has been added.
Response 2: Thank you for pointing out the problem, the key words have been added to the end of the abstract.
Response 3: Thank you for your suggestion, "wt%" has been added in the article.
Response 4: Thank you for pointing out the problem, this paragraph has been deleted from the article.
Response 5: Thank you for pointing out the problem, the punctuation has been changed in the article.
Response 6: Thank you for your suggestions. The reference format of the article has been modified.
Please refer to the attachment to revise the position of the latest manuscript, thank you!

Reviewer 3 Report
Comments and Suggestions for Authors
This work presents a method for deep sulfurization of petroleum coke containing over 7% wt. sulfur. It is a process in two stages: the first stage is alkali roasting, and the second stage consists in oxidation with H2O2 in presence of acetic acid and assisted by ultrasonic power. The results of this process are remarkable, and they are demonstrated through different modern analysis methods.
The Abstract includes the main information about the experimental study and the main results. It can stand alone for the work achievement. The Keywords were chosen rightly. The Introduction is comprehensive, framing the work in the recent literature. The Materials and Methods are concisely described. The presentation of the results is clear and perfectly understandable. The discussion of the results is detailed and with correct interpretations. Conclusions are well and concisely formulated.
The authors are requested to address to the following observations:
1. In Introduction, in lines 50-52, you claim that the nitrogen oxides and sulfur oxides are greenhouse gases, but the main impact on the environment is that they cause acid rain and are toxic for human beings and animals. Only nitrous oxide N2O is acknowledged for its greenhouse effect. Please correct the statement.
2. A little mess was produced with the citations in lines 75-77: no reference is given for the solvent extraction desulfurization (in fact, it is reference [25], wrongly cited at hydrodesulfurization); reference [21] is wrongly cited at alkali metal desulfurization (in fact, it should be cited at oxidative desulfurization).
3. In section 2.1. Materials and Methods there is a phrase which seems misplaced, in lines 151-153: “ Interventionary studies involving animals and humans…ethically approved code”.
4. In section 2.2. Experimental methods, the quantity of the coke taken in a batch in the first stage and then, more important, in the second stage should be disclosed. The authors say that they used 1 to 10 mL acetic acid (relative to what quantity of coke?)
5. In Section 3.1. you proposed some reaction mechanisms. The Equations 2 and 3 suffer from lack of stoichiometry. The sulfur in coke is found mainly in thiophenes (R-C4H4S), organic sulfides (R-S-R1) and mercaptans (R-SH), where R and R1 are aliphatic moieties.
More probably, the mechanism for mercaptans, is the following: the first step is the thermal decomposition of the mercaptans resulting in olefin and hydrogen sulfide. Then H2S react instantly with NaOH forming Na2S and water:
2 R-SH →R’ +H2S ; H2S+ 2 NaOH→Na2S+2 H2O (2)
The mechanism for thiophenes and organic sulfides can be considered alike, since the sulfur atom is bonded similarly in the molecule: -S- . Your proposed mechanism is possible considering the products detected by X-rays diffraction, but stoichiometry is unbalanced. You could write:
7 (-S-)+6 NaOH→ 2Na2S + Na2SO4+3H2S+ SO2 (3)
You could take into consideration (not compulsory) my suggestions at this point. These modifications wouldn’t affect the meaning of Figure 3.
6. In my opinion, the biggest lack of the article is that authors did not calculate the yield of the desulfurized coke. Normally, they should weight the coke before and after each stage of the process and calculate the mass loss/ desulfurized coke yield. Since usually the yields are low, this may be an important economic concern.
Author Response
Dear reviewer,
Thank you for your comments on the article, the responses to your questions are posted below:
Response 1: The sentence after polishing: "Thank you for bringing this matter to our attention. We have replaced the term 'greenhouse gases' with 'harmful gases.'
Response 2: The polished sentence: "I regret the citation error in my paper, which was due to my oversight. Thank you for bringing it to my attention, and I have made the necessary revisions."
Response 3: The polished sentence: "Thank you for bringing the error to my attention. I have implemented the necessary revisions."
Response 4: Thank you for your correction. The inclusion of "4g of petroleum coke" has been made in the experimental methodology section of the article.
Response 5: Thank you for your valuable suggestions. I think the equation you proposed to be an improvement over my original one, and I have completed the modifications.
Response 6: I would like to express my sincere gratitude for your pointing out the shortcomings in the paper. Since the reduction in petroleum coke quality was found to be related to the calcination temperature, the addition of alkali, and the calcination time during the experiment, the reduction could be due to the decrease in volatiles or the loss of carbon in petroleum coke due to the action of alkali. However, it is unclear whether the reduction of petroleum coke is positive or negative, so the paper does not include the yield of petroleum coke. Your suggestion is very meaningful. In the follow-up research work, the yield and fixed carbon content of petroleum coke products should be taken into account while desulphurizing petroleum coke.
Please refer to the attachment to revise the position of the latest manuscript, thank you!
